# Development and Characterization of Inhaled Ethanol as a Novel Pharmacological Strategy Currently Evaluated in a Phase II Clinical Trial for Early-Stage SARS-CoV-2 Infection

**DOI:** 10.3390/pharmaceutics13030342

**Published:** 2021-03-05

**Authors:** Ana Castro-Balado, Cristina Mondelo-García, Letricia Barbosa-Pereira, Iria Varela-Rey, Ignacio Novo-Veleiro, Néstor Vázquez-Agra, José Ramón Antúnez-López, Enrique José Bandín-Vilar, Raquel Sendón-García, Manuel Busto-Iglesias, Ana Rodríguez-Bernaldo de Quirós, Laura García-Quintanilla, Miguel González-Barcia, Irene Zarra-Ferro, Francisco J. Otero-Espinar, David Rey-Bretal, José Ramón Lago-Quinteiro, Luis Valdés-Cuadrado, Carlos Rábade-Castedo, María Carmen del Río-Garma, Carlos Crespo-Diz, Olga Delgado-Sánchez, Pablo Aguiar, Gema Barbeito-Castiñeiras, María Luisa Pérez del Molino-Bernal, Rocío Trastoy-Pena, Rossana Passannante, Jordi Llop, Antonio Pose-Reino, Anxo Fernández-Ferreiro

**Affiliations:** 1Pharmacy Department, University Clinical Hospital of Santiago de Compostela (SERGAS), 15706 Santiago de Compostela, Spain; ana.castro.balado@gmail.com (A.C.-B.); crismondelo1@gmail.com (C.M.-G.); iria.varela.rey@sergas.es (I.V.-R.); enrique.jose.bandin.vilar@sergas.es (E.J.B.-V.); manuel.busto.iglesias@sergas.es (M.B.-I.); laura.garcia.quintanilla@sergas.es (L.G.-Q.); miguel.gonzalez.barcia@sergas.es (M.G.-B.); irene.zarra.ferro@sergas.es (I.Z.-F.); 2Clinical Pharmacology Group, Health Research Institute of Santiago de Compostela (IDIS), 15706 Santiago de Compostela, Spain; 3Pharmacology, Pharmacy and Pharmaceutical Technology Department, Faculty of Pharmacy, University of Santiago de Compostela (USC), 15782 Santiago de Compostela, Spain; francisco.otero@usc.es; 4Analytical Chemistry, Nutrition and Food Science Department, Faculty of Pharmacy, University of Santiago de Compostela (USC), 15782 Santiago de Compostela, Spain; letricia.barbosa.pereira@usc.es (L.B.-P.); raquel.sendon@usc.es (R.S.-G.); ana.rodriguez.bernaldo@usc.es (A.R.-B.d.Q.); 5Internal Medicine Department, University Clinical Hospital of Santiago de Compostela (SERGAS), 15706 Santiago de Compostela, Spain; ignacio.novo.veleiro@sergas.es (I.N.-V.); nestor.vazquez.agra@sergas.es (N.V.-A.); 6Pathological Anatomy Department, University Clinical Hospital of Santiago de Compostela (SERGAS), 15706 Santiago de Compostela, Spain; jose.ramon.antunez.lopez@sergas.es; 7Molecular Image Group, Health Research Institute of Santiago de Compostela (IDIS), 15706 Santiago de Compostela, Spain; davidrey.bretal@usc.es (D.R.-B.); pablo.aguiar.fernandez@sergas.es (P.A.); 8Pneumology Department, University Clinical Hospital of Santiago de Compostela (SERGAS), 15706 Santiago de Compostela, Spain; jose.ramon.lago.quinteiro@sergas.es (J.R.L.-Q.); luis.valdes.cuadrado@sergas.es (L.V.-C.); carlos.rabade.castedo@sergas.es (C.R.-C.); 9Clinical Analytic Department, University Clinical Hospital of Santiago de Compostela (SERGAS), 15706 Santiago de Compostela, Spain; maria.del.carmen.del.rio.garma@sergas.es; 10Pharmacy Department, University Clinical Hospital of Pontevedra (SERGAS), 36162 Pontevedra, Spain; carlos.crespo.diz@sergas.es; 11Sociedad Española de Farmacia Hospitalaria (SEFH), 28001 Madrid, Spain; olga.delgado@ssib.es; 12Microbiology Department, University Clinical Hospital of Santiago de Compostela (SERGAS), 15706 Santiago de Compostela, Spain; gema.barbeito.castineiras@sergas.es (G.B.-C.); maria.luisa.perez.del.molino.bernal@sergas.es (M.L.P.d.M.-B.); rocio.trastoy.pena@sergas.es (R.T.-P.); 13Radiochemistry Department, CIC biomaGUNE, Parque Tecnológico de San Sebastian, 20009 San Sebastián, Spain; rpassannante@cicbiomagune.es

**Keywords:** COVID-19, SARS-CoV-2, inhaled ethanol, molecular imaging, PET

## Abstract

Inhaled administration of ethanol in the early stages of COVID-19 would favor its location on the initial replication sites, being able to reduce the progression of the disease and improving its prognosis. Before evaluating the efficacy and safety of this novel therapeutic strategy in humans, its characterization is required. The developed 65° ethanol formulation is stable at room temperature and protected from light for 15 days, maintaining its physicochemical and microbiological properties. Two oxygen flows have been tested for its administration (2 and 3 L/min) using an automated headspace gas chromatographic analysis technique (HS-GC-MS), with that of 2 L/min being the most appropriate one, ensuring the inhalation of an ethanol daily dose of 33.6 ± 3.6 mg/min and achieving more stable concentrations during the entire treatment (45 min). Under these conditions of administration, the formulation has proven to be safe, based on histological studies of the respiratory tracts and lungs of rats. On the other hand, these results are accompanied by the first preclinical molecular imaging study with radiolabeled ethanol administered by this route. The current ethanol formulation has received approval from the Spanish Agency of Medicines and Medical Devices for a phase II clinical trial for early-stage COVID-19 patients, which is currently in the recruitment phase (ALCOVID-19; EudraCT number: 2020-001760-29).

## 1. Introduction

Coronaviruses (CoVs) are single-stranded RNA viruses which can infect animals and humans, causing respiratory, gastrointestinal, hepatic and neurologic diseases [1]. In December 2019, several health authorities reported patients with pneumonia of an unknown cause, which were epidemiologically linked to a seafood market in Wuhan, China. The pathogen, a novel coronavirus (SARS-CoV-2), was identified by local hospitals and the infection was called coronavirus infectious disease 2019 (COVID-19) [2,3]. In March 2020, the World Health Organization (WHO) classified COVID-19 as a pandemic, and in January 2021, a year after its eruption, there have been more than 93 million confirmed cases and two million deaths [4].

Since the outbreak of the COVID-19 pandemic, it has been seen that compliance of control measures such as physical distancing, the use of masks, hand cleaning, tracing contacts, testing of exposed or symptomatic persons and isolation have restricted transmissions [5]. Even so, these actions have not been implemented uniformly, and they have not shown to be enough to prevent the spread of SARS-CoV-2.

Vaccines are needed to produce group immunity and reduce COVID-19 morbidity and mortality. In this sense, several vaccine platforms have been implicated in the rapid development of candidate vaccines [6,7,8], which allowed the starting of the vaccination process worldwide in December 2020.

According to the pathological characteristics of COVID-19, especially for patients with moderate to severe COVID-19, several treatment strategies have been developed, including, among others, antiviral agents, inflammation inhibitors/antirheumatic drugs and low molecular weight heparins [9,10]. Concerning new therapies, it is necessary to highlight the administration of convalescent plasma with high IgG titers against SARS-CoV-2, which have revealed promising results, although further studies are required [11]. In addition, regarding antivirals, plitidepsin has demonstrated potent preclinical efficacy against SARS-CoV-2 by targeting the host protein eEF1A, and it is being tested in a proof of concept study to evaluate its safety profile (NCT04382066) [12]. Another possible alternative is treatment with monoclonal antibodies, such as bamlanivimab, which is currently evaluated in an expanded access program (NCT04603651) [13], or the cocktail of monoclonal antibodies anti-spike SARS-CoV-2, which is being tested in ambulatory adult and pediatric patients (NCT04425629) [14].

Despite all the above, it has been seen that, in the early stages, antiviral drugs can prevent the progression of the disease, while dexamethasone [15], immunomodulators and antiviral drugs seem to improve the clinical outcome of patients with severe COVID-19 [9]. However, treatment remains elusive in early stages, in which there are few strategies that can bring benefits; some strategies have failed, and others are under evaluation [11]. 

In pandemic situations like this, a significant number of patients find themselves in a therapeutic vacuum, without effective drugs to address their treatment. In this context, drug repositioning is a strategy to generate additional value from an approved drug, using it for a different therapeutic purpose than that for which it was originally intended [16]. The absence of evidence-based treatments for COVID-19 has led to the start of a large number of clinical trials in order to offer patients the most effective and safest therapeutic options. In this sense, the genetic characteristics of SARS-CoV and MERS-CoV suggest that SARS-CoV-2 may be susceptible to disinfectants such as ethanol, with a graduation between 62–71°, with proven activity against enveloped viruses [17,18].

In early stages of the disease, active viruses are located in the throat and lungs. This way, the administration of viricidal agents in the initial replication place could decrease viremia in the first stages of the disease, and consequently reduce its progression and improve the prognosis drastically [19]. In hospital pharmacy departments, it is common to elaborate ethanol formulations for their use in non-usual routes of administration. Ethanol was used in catheter seals to prevent bacterial growth [20], as a neurolytic in the peripheral or central nervous block in terminally ill patients [21], in esophageal varicose sclerotherapy [22], hemorrhage control in hepatocellular carcinoma surgeries [23], debridement of the corneal epithelium [24] or even as an intravenous antidote in the treatment of ethylene glycol or methanol poisoning [25].

COVID-19 causes a particularly severe illness in older adults. The percentage of hospitalized patients within this age group is high [11], and over 95% of total death cases occurred in people older than 60 years, with more than 50% of all deaths being people aged 80 years or older [3,26,27]. In this situation, with no alternatives among commercial medicines, it is necessary to develop new therapeutic approaches which can be an adequate option in elderly patients. In this sense, the local administration of ethanol could be effective against a viral envelope with no systemic adverse effects [28,29].

This study presents a novel pharmacological strategy against SARS-CoV-2 with inhaled ethanol and its galenic, toxicological and pharmacokinetic characterization. The current ethanol formulation has received approval from the Spanish Agency of Medicines and Medical Devices to test its efficacy-safety in a phase II clinical trial in elderly patients with COVID-19, which is currently in the recruitment phase (ALCOVID-19; EudraCT number: 2020-001760-29) [30].

## 2. Materials and Methods

This work includes an initial galenic characterization phase, in which the physiochemical and microbiological stability of the formulation in humidifying flasks was determined, as well as the concentration of vaporized ethanol through two oxygen flows. Subsequently, a two-part preclinical phase was carried out. On the one hand, an in vivo study to ensure the safety of the ethanol administered by this route was performed in Sprague-Dawley rats. On the other hand, evaporated ethanol was radiolabeled, and its pharmacokinetics in rats were studied using molecular imaging techniques with positron emission tomography/computed tomography (PET/TC). Lastly, a preliminary clinical study was carried out in six healthy volunteers divided into two groups (1:1), who were subjected to the administration of 15 minutes of oxygen therapy at 2 L/min and 3 L/min flow through a Ventimask^®^
(Flexicare, Mountain Ash, UK) face mask. These data are depicted in the Appendix A.

### 2.1. Preparation of the 65° Ethanol Solution and Stability Determination in Disposable Humidifying Bottles

A volume of 250 mL of 65° ethanol solution was packaged in disposable humidifier bottles (INTL CE0482. Ref. 3230, generously donated by Oximesa Nippon Gases, Madrid, Spain). To prepare, the starting point was 163 mL of 99.5° ethanol with PhEur indication on its label(PanReac AppliChem^®^, Darmstadt, Germany), which was measured using a graduated cylinder, and completed with sterile water (Fresnius Kabi^®^, Barcelona, Spain) up to 250 mL. After homogenization, it was necessary to wait until it reached room temperature to measure ethanol graduation with a 60–70° Gay-Lussac alcoholmeter (Boeco^®^, Hamburg, Germany). Sterilizing vacuum filtration through a 0.22-micron filter was performed in a horizontal laminar flow cabinet.

In order to determine the stability of ethanol over 15 days in a disposable humidifying bottle, an initial ethanol graduation measurement was carried out, and then once weekly using the 60–70° Gay-Lussac alcoholmeter. During this period, the ethanol solution was kept in a fully sealed humidifier bottle at room temperature, and protected from light. Parallel to this, a microbiological study was carried out by extraction of three 3 mL samples: one sample before sterilizing filtration and another after this process on day 0; and the last one after 15 days of storage at room temperature and protected from light. These samples were cultivated in a thioglycollate broth (Merck^®^, Darmstadt, Germany), Columbia blood agar (Merck^®^, Darmstadt, Germany) and Sabouraud (Merck^®^, Darmstadt, Germany). All mediums were incubated aerobically at 37 °C; thioglycollate for 10 days; and blood and Sabouraud agar plates for 48 h. Sabouraud agar plates were subsequently incubated for 13 days in aerobiosis at room temperature.

### 2.2. Determination of Ethanol in the Administered Oxygen Flow

A flow of oxygen was made to pass through a humidifier containing the 65° ethanol solution. The effect of two oxygen flows was compared: 2 and 3 L/min. These flows cause the evaporation of the ethanol in the humidifier. This determination of evaporated ethanol will allow choosing the optimal oxygen flow that causes the evaporation without generating aerosols, with this being best suited to its use later in the clinical trial.

The ethanol quantification has been carried out by an automated headspace gas chromatographic mass spectrometry analysis (HS-GC-MS) (Figure 1). It was performed using a Finnigan Trace GC Ultra chromatograph coupled with a Finnigan Trace DSQ mass detector, equipped with a Thermo Scientific Head Space (HS) TriPlus autosampler (Thermo Fisher Scientific, San José, CA, USA). Sampling was performed in 21.5 mL HS glass vials, which were previously sealed. The septum was perforated with two needles to allow a way in and a way out for the gas. Then, oxygen enriched in ethanol generated in the humidifier was flushed for 2 min in each vial to replace all of its internal air, and guarantee that the sample was representative. The samples were collected at several intervals of time, as follows: 0–2 min; 2–4 min; 6–8 min; 13–15 min; 28–30 min; and 43–45 min, in order to determine the concentration of ethanol at 2, 4, 8, 15, 30 and 45 min of administration, respectively. All experiments were performed in triplicate for the two flow rates tested (2 L/min and 3 L/min). The vials were maintained and refrigerated before HS-GC-MS analysis.

Samples were incubated at 45 °C for 4 min, before 0.2 mL of headspace gas was withdrawn by the autosampler, using a 2.5 mL syringe heated at 50 °C. The gas volume was injected into the inlet of the Trace GC set at 175 °C in a constant temperature split mode, at a split ratio of 50:1 and the split flow of 50 mL/min. The analytical column was a Rxi-624Sil MS, 30 m in length, with an internal diameter of 0.25 mm (1.4 μm film thickness) from Restek (Bellefonte, PA, USA). Helium was used as carrier gas at a flow rate of 1.0 mL/min in a constant flow mode. The oven temperature was initially set at 40 °C for 4 min, then increased at a rate of 50 °C/min until 120 °C, and held at 120 °C for 2.5 min. The total runtime of each analysis was 10 min. The temperatures of the GC transfer line and the ion source of the MS were set at 220 °C and 200 °C, respectively. The mass spectrometer was operated in an electron impact ionization (EI+) mode, and the chromatograms were acquired in full scan mode over the m/z range of 20–150, from 2 to 10 min, at a scan rate of 3.55 scans/s. The chromatographic peak of ethanol was identified by comparison with reference spectra from the NIST 2011/Wiley 9 Combined Mass Spectral Library, using NIST Mass Spectral Search Program (version 2.0), and confirmed by comparison of its retention time (3.4 min), with that obtained by analysis of authentic standard under the same conditions. The instrument control and data acquisition and processing were performed with Xcalibur 2.0.7 software (Thermo Fisher Scientific Inc., Waltham, MA, USA).

The calibration curve of ethanol was prepared in 21.5 mL HS glass vials, which were previously sealed with Teflon septum caps, by adding different volumes of 99.5% (0.5–2 µL) ethanol to reach the headspace (vapor phase) concentrations in the range of 20–100 mg/L. The equation and determination coefficient obtained from three replicates were y = 82.285x + 9.8808 and R^2^ = 0.9993, respectively.

### 2.3. Preclinical Studies

#### 2.3.1. Ethanol Exposure Toxicological Studies and Immunohistochemical Analysis

These studies were carried out on six female Sprague-Dawley rats (four receiving inhaled ethanol and two control rats). They were supplied by the animal facilities at the University of Santiago de Compostela, and the average weight was 250 ± 25 g. During the experiments, animals were kept in individual cages under a controlled temperature (22 ± 1 °C) and humidity (60 ± 5%) conditions, with day-night cycles regulated by artificial light (12/12 h) and fed *ad libitum*. All animal experiments complied with the ARRIVE guidelines [31] and were carried out in accordance with the EU Directive 2010/63/EU for animal experiments, being approved by the Galician Network Committee for Ethics Research. All experimental procedures were approved by the ethical committee and the local authorities before conducting experimental work.

For the administration, 65° ethanol was placed in a disposable humidifier and vaporized by passing an oxygen flow of 2 L/min. The generated ethanol vapors went through a tube to an inhalation chamber of 20 × 20 × 25 cm (Bioseb, FL, USA) (Figure 2). For exposure to the ethanol vapours, each rat was placed in the chamber for 15 min every 8 h (three times a day) for five consecutive days. Following the exposure time, each rat was removed from the inhalation chamber and returned to its individual cage.

The animals were sacrificed by an intracardiac injection of 5 mL of potassium chloride (1 mEq/mL; B. Braun Medical, S.A, Barcelona, Spain) at day +6 of the initiation assay. Upper respiratory tracts were removed and individually fixed in 10% formalin, dehydrated, paraffin embedded, sectioned in slices with 4 μm thickness, and stained with H&E (haematoxylin and eosin). The samples were blindly evaluated by a lung pathologist specialist using a microscope (Zeiss, Oberkochen, Germany). Sections were examined using light microscopy, and digital images were acquired using Leica^®^ software (Leica Microsystems, Wetzlar, Alemania). Analyses were performed at a final magnification of ×1000. Five nonoverlapping fields of view per section from two to three sections (from different regions of the lung, esophagus and trachea) per animal were analyzed.

#### 2.3.2. Preclinical Pharmacokinetics

The preclinical pharmacokinetic studies were carried out with positron emission tomography/computed tomography (PET/TC) methodology using radiolabeled ethanol. The use of molecular imaging, particularly PET/TC, provides a non-invasive imaging technique that visualizes the distribution of different radiotracers over time in animal models [32]. In this study, 1-^11^C-ethanol has been synthetized in order to know the distribution of ethanol administered into the respiratory tract in rats along time, in order to assess the tolerance and safety of the administration of vaporized ethanol three times a day for five days in Sprague-Dawley rats.

##### Synthesis of 1-^11^C-Ethanol

The synthesis of 1-^11^C-ethanol was carried out using a TRACERlab FXC Pro synthesis module (GE Healthcare, Chicago, IL, USA). [^11^C]CO_2_ was generated in an IBA Cyclone 18/9 cyclotron (IBA RadioPharma Solutions Headquarter, Louvain-la-Neuve, Belgium) by proton irradiation (target current = 22 µA, integrated current = 2 µAh) of a gas N_2_/O_2_ mixture (99/1, starting pressure = 20 bar) with high energy (18 MeV) protons. The radioactive gas was first trapped in a molecular sieve oven at room temperature and then released by heating at 180 °C under nitrogen flow (20 mL/min). The released [^11^C]CO_2_ was bubbled in a reaction vial containing CH_3_MgBr (1M solution in THF, 250 µL, Merck^®^, Darmstadt, Germany). After complete trapping, LiAlH_4_ (1M solution in THF, 500 µL, Merck^®^, Darmstadt, Germany) was added, and the resulting mixture was stirred at 80 °C for 5 min. The solvent was then evaporated under helium flow (5 min, 60 °C; then 1 min, 80 °C). The reactor was cooled to 40 °C and aqueous HCl (4 M, 1 mL) was immediately added. The solution was stirred for 30 s, filtered using a 0.2 µm filter, and purified by high-performance liquid chromatography (HPLC) using a Mediterranea SEA18 (250 × 10 mm, 5 µm particle size) (Teknokroma Analítica, Barcelona, Spain) column as a stationary phase and ultrapure water as the mobile phase (flow rate = 5 mL/min). The purified product (retention time = 7.7 min; total collected volume of ca. 2 mL) was collected in a vial and diluted 1:1 with a physiologic saline solution. The amount of radioactivity of the final radiotracer was measured in a dose calibrator (PETDOSE HC, Comecer, Castel Bolognese RA, Italy). Radiochemical purity was determined by HPLC, using an Agilent 1200 Series HPLC system (Agilent Technologies, Santa Clara, CA, USA) with a multiple wavelength UV detector (λ = 254 nm) and a radiometric detector (Raytest, Elysia-raytest GmbH, Straubenhardt, Germany). A Tracer Excel 120 C8 (250 × 4.6 mm, 5 µm particle size) (Teknokroma Analítica, Barcelona, Spain) was used as stationary phase and purified water as mobile phase (flow rate = 1 mL/min; retention time = 3.95 min).

##### Lung PET Studies

Three healthy female Sprague-Dawley rats (*n* = 3) weighting 350 ± 10 g were used in this study. For PET studies, anesthesia was induced with 5% isoflurane and maintained by 1.5 to 2% of isoflurane in 100% O_2_. The labelled compound (ca. 7 MBq, 100 μL) was administered through endotracheal insufflations using the PennCentury MicroSprayer^®^ Aerosolizer (FMJ-250 High Pressure Syringe Model, Penn-Century, Inc., Wyndmoor, PA, USA) and a small animal laryngoscope (Penn-Century, Model LS-2) (Penn-Century. Inc. Wyndmoor, USA) for the correct visualization of the epiglottis. After administration, the animal was quickly moved into the PET camera, and the PET acquisition began. The time gap between administration of the dose and start of image acquisition was 1 min. During imaging, rats were kept normothermic using a heating blanket (Homeothermic Blanket Control Unit; Bruker, MA, USA). PET Imaging was performed using an eXploreVista-CT small animal PET-CT system (GE Healthcare, Chicago, IL, USA). Whole body dynamic images were acquired in four bed positions (20 frames: 4 × 5 s, 4 × 15 s, 4 × 30 s, 4 × 60 s, 4 × 120 s; total acquisition time = 61.33 min) in the 400–700 keV energetic window. After each PET acquisition, a CT scan (X-ray energy: 40 kV, intensity: 140 μA) was performed for a later attenuation correction in the image reconstruction, as well as for the unequivocal localization of the radioactivity. Random and scatter corrections were also applied to the reconstructed image. PET-CT images of the same animal were co-registered and analyzed using the PMOD image processing tool (PMOD Technologies Ltd., Zürich, Switzerland). Volumes of interest (VOIs) were placed on major organs (lungs, liver, kidneys, heart and brain), and time–activity curves (decay corrected) were obtained as cps/cm^3^ in each organ. Curves were transformed into real activity (Bq/cm^3^) curves by using a calibration factor, obtained from previous scans performed on a phantom (micro-deluxe, Data spectrum Corp., Durham, NC, USA) under the same experimental conditions (isotope, reconstruction algorithm and energetic window).

## 3. Results

### 3.1. Stability of the 65° Ethanol Pharmaceutical Compounding and Flow Oxygen Effect

The graduation of the humidifying bottle with the hydro-alcoholic solution under storage conditions (without use) remained stable for 15 days, with no microbiological growth observed at the end of said period.

The formulation maintains optimal concentrations (decreases less than 5% of the initial concentration) at the end of the three daily applications, with both flows tested (2 and 3 L/min). With the flow of 2 L/min, a solution graduation of 64.1° was maintained (98.2% of the original concentration of the hydro-alcoholic mixture), while at the flow of 3 L/min, the graduation obtained was 63.3° (97.8%). The ethanol content of the initial solution, in the humidifier bottle, decreased 4.63 g at 2 L/min and 5.38 g at 3 L/min, after 45 min of administration. The difference between flow rates was 0.75 g of ethanol emitted per day, higher for a flow of 3 L/min, which represents 16% more than the total amount achieved with the flow of 2 L/min, as shown in Figure 3a.

Considering the minute ventilation of 5 L/min for human adults recommended for short-term exposure [33], and the inhalation:exhalation (I:E) ratio of 1:2, the total amount of ethanol inhaled after the three daily administrations at 2 L/min and 3 L/min was 1.51 g and 1.79 g, respectively (Figure 3b). The difference in the total amount of the inhaled ethanol between flow rates was 0.28 g higher for the highest flow rate used.

Kinetic parameters of the ethanol vaporization in the oxygen stream, for both flow rates tested, are shown in Figure 4. Second-degree polynomial equations were developed from the experimental data, and areas under the concentration-time curves (AUC) were calculated to determine the amount of ethanol emitted during the treatments. At the flow rate of 2 L/min, the average concentrations of ethanol yielded (51.5 ± 4.5 mg/L of oxygen (8.7% CV)) were higher than that observed at 3 L/min (39.9 ± 5.7 mg/L of oxygen (14.4% CV)) during the 45 min of treatment. The concentration of ethanol decreased slightly during the time for both flow rates tested. In the first 15 min, the average concentrations of ethanol determined at a flow rate of 2 L/min and 3 L/min were 56.6 ± 1.9 (3.4% CV) and 46.6 ± 3.1 mg/L (6.6% CV), respectively. In the second period (15–30 min) of the experiment, the average concentrations were 51.0 ± 1.6 (3.2% CV) and 38.6 ± 2.0 mg/L (5.2% CV), and finally, in the last period of 15 min, the concentrations of ethanol were 46.5 ± 1.46 (2.9% CV) and 34.0 ± 1.0 mg/L (3.0% CV), for 2 L/min and 3 L/min, respectively.

Following the previous consideration for the I:E ratio, the average concentrations of inhaled ethanol at flow rates of 2 L/min and 3 L/min were 6.70 ± 0.75 mg/L/min (11.2% CV) and 7.99 ± 1.12 mg/L/min (14.7% CV), respectively. The concentrations were slightly higher for the highest flow rate, since the dilution with the assistant air was lower. Specifically, the inhaled concentrations (mg/L/min) at the different administrations were: 7.55 ± 0.27 (0–15min), 6.70 ± 0.27 (15–30 min) and 5.58 ± 0.26 (30–45 min) at the flow rate of 2 L/min; and 9.37 ± 0.62 (0–15 min), 7.76 ± 0.41 (15–30 min), 6.81 ± 0.19 (30–45 min) at the flow rate of 3 L/min.

Taking into account the total amount of ethanol emitted per minute of treatment (mg/min), the values were lower for the flow rate of 2 L/min (100.1 ± 11.5 mg/min (11.5% CV)) than the 3 L/min (average of 119.7 ± 17.1 mg/min (14.3% CV)), as shown in Figure 5. These differences were higher in the first minutes of ethanol enriched oxygen administration, with total amounts of ethanol of 112.7 ± 4.1 (3.7% CV) for 2 L/min flow rate and 139.7 ± 9.2 (6.6% CV) for 3 L/min, and decreased during the time of around 23% and 27%, respectively. For the second and third administrations at 2 L/min flow rate, the amounts of ethanol were 99.7 ± 4.2 (4.2% CV) and 87.4 ± 4.2 (4.2% CV), respectively, whereas at a flow rate of 3 L/min, the values observed were 115.8 ± 6.0 (5.2% CV) and 102.6 ± 3.0 (2.9% CV). 

Therefore, based on previous considerations, the estimated final average amounts of inhaled ethanol (mg/min) were 33.6 ± 3.6 (10.7% CV) and 40.0 ± 5.7 (14.2% CV) at the flow rate of 2 L/min and 3 L/min, respectively. At the flow rate of 2 L/min, analysis of variance (ANOVA) showed no significant differences (*p* > 0.05) among the doses of inhaled ethanol during the three administrations: 37.60 ± 2.04 (0–15 min); 33.36 ± 1.34 (15–30 min); and 29.31 ± 1.32 (30–45 min). On the contrary, significant differences (*p* < 0.001) were observed at the flow rate of 3 L/min: 46.55 ± 1.02 (0–15 min); 38.57 ± 2.00 (15–30 min); and 34.04 ± 1.01 (30–45 min).

### 3.2. Preclinical Studies

#### 3.2.1. Ethanol Exposure Toxicological Studies and Immunohistochemical Analysis

Tissue and air-space fractions (atelectasis), oedema and congestion were evaluated in the paraffin-embedded sections of lung tissue stained with hematoxylin and eosin. There were no typical patterns of pulmonary toxicities along the respiratory tract, as can be seen in Figure 6 (a. lung; b. and c. trachea and d. and e. esophagus).

#### 3.2.2. Pharmacokinetics

##### Synthesis of 1-^11^C-Ethanol

The synthesis of 1-^11^C-ethanol has been previously reported [34,35]. In these previous works, the final purification of the labeled compound was achieved by fractional distillation. In our hands, this synthetic approach resulted in the presence of radioactive impurities in the final solution. The undesired by-products were co-eluted with methanol (retention time = 3.5 min) and isopropanol (retention time = 4.7 min) under our analytical HPLC conditions (Figure 7a) and accounted for ca. 20% of total radioactivity. Taking into account that our aim was to obtain radiochemically pure 1-^11^C-ethanol, and that the overall yield was not a critical aspect, we decided to assay a purification method based on HPLC. After a filtration step to remove any eventual precipitate due to incomplete dissolution of the solid residue after the addition of hydrochloric acid, a good separation of the different peaks could be achieved when water was used as the mobile phase.

Under these conditions, approximately 600 MBq of pure 1-^11^C-ethanol (radiochemical purity > 99%; Figure 7b) could be obtained in average production time of 30 min (decay corrected radiochemical yield of around 7%). This amount of radioactivity, which was ready for administration after simple dilution with physiologic saline solution, was sufficient to tackle subsequent in vivo studies in rodents (see below).

##### Lung PET Studies

PET studies were carried out to determine the biodistribution of 1-^11^C-ethanol in rats after intratracheal insufflation. Administration was carried out using the Penn-Century MicroSprayer^®^ Aerosolizer, which is reported to provide around a 20 µm droplet size [36]. According to our previous results, this is a very appropriate system for the quantitative administration of aerosols in the rat lung [36]. Due to the administration process, dynamic images could be started around one minute after administration, and hence initial distribution data (0.1 min after administration) was lost. Dynamic PET images (Figure 8) show a very fast lung clearance (*t*_1/2_ = 1.43 min), with most of the radioactivity already cleared from the lungs at *t* = 1 min. 

The presence of radioactivity delocalized over the whole animal at early time points suggest translocation to the blood, followed by progressive accumulation in the liver (Figure 8), suggesting metabolic oxidation to acetaldehyde by alcohol dehydrogenase and cytochrome P450, which are extensively present in this organ.

## 4. Discussion

The emergency period of the COVID-19 outbreak has forced the scientific community to generate evidence against the clock [37]. Despite the recent development of vaccines to prevent the disease, new therapeutic alternatives to treat the established disease should be studied [38,39]. According to the pathological characteristics of COVID-19 and different clinical stages, especially for patients with moderate to severe disease, antiviral agents, inflammation inhibitors/antirheumatic drugs, low molecular weight heparins and convalescent plasma with high IgG titers against SARS-CoV-2 have been used and tested [9,10,11]. In the early stages of the disease, the treatment of COVID-19 remains elusive. There are few strategies that can bring benefits, some strategies have failed, and others are under evaluation [11,40,41,42,43].

Faced with therapeutic gaps such as these, clinicians are forced to resort to therapeutic alternatives such as drug repositioning and pharmaceutical compounding. The genetic characteristics of SARS-CoV and MERS-CoV suggest that SARS-CoV-2 may be susceptible to disinfectants with proven activity against enveloped viruses [17,18]. Ethanol exerts its action against a viral envelope; a lipid bilayer taken from the host cells in the assembly or budding stage of the viral cycle, causing viral lysis with the consequent release and degradation of its content [19]. Active viruses in throat and lungs were isolated in patients with a mild condition only up to day 8 after the onset of symptoms, reaching the peak of the viral load before day 5. This way, the administration of viricidal agents in the place of initial replication could decrease viremia in the first stages of the disease, and consequently reduce the progression of the disease and improve its prognosis [29,44]. For this reason, the administration of inhaled ethanol could be presented as a new therapeutic strategy to prevent the progression of COVID-19 infections [45]. Our hypothesis of treatment with inhaled ethanol focuses on its use in institutionalized elderly patients, since certain therapies used to date were contraindicated or discouraged in this age group, where COVID-19 has severe outcomes with a high mortality rate [46]. Prior to the start of a phase II clinical trial for early-stage COVID-19 older adult patients, already approved by the Spanish Agency of Medicines and Medical Devices (ALCOVID-19; EudraCT number 2020-001760-29), galenic, toxicological and pharmacokinetic characterization of the inhaled ethanol compounded formulation has been necessary.

After the vaporization of 65° ethanol for 45 minutes in three administrations, there is a decrease in the alcohol content of the hydro-alcoholic mixture contained in the disposable humidifiers compared to the initial values. Through bubbling oxygen at different flows through the 65° ethanol solution, evaporation of the mixture is favored. The vapor pressure of ethanol is much higher than the vapor pressure of water, since ethanol has a boiling point (78.4 °C) considerably lower than water (100 °C) [47]. For this reason, the gas produced after bubbling oxygen through the 65° ethanol solution will have a higher ethanol concentration than the starting mixture.

Considering the minute ventilation of 5 L/min for human adults recommended for short-term exposure [33], and the I:E ratio of 1:2, the average concentrations of inhaled ethanol in the three administrations of ethanol were 6.70 mg/L/min and 7.99 mg/L/min for the flow rates of 2 L/min and 3 L/min, respectively, which correspond to the total amounts of ethanol per minute of 33.52 mg/min and 39.88 mg/min. These amounts were lower than those observed by Bessonneau and Thomas [48] in a study related to the exposure after hand disinfection, where the total inhaled dose of ethanol ranged between 150.8 and 219.26 mg/min. The total amount of alcohol vaporized during the flushing of oxygen through the formulation increased with the flow rate of oxygen, as observed by other studies described in the literature [49]. The concentrations of ethanol per min (mg/L/min) emitted in the humidifier were lower at the flow rate of 3 L/min (see Figure 4), as the higher flow rate of oxygen could not reach the equilibrium with ethanol at the same time and carried less ethanol vapor per liter (mg/L). The administration during the time of treatment was more stable at the flow rate of 2 L/min than 3 L/min (see Figure 4).

The total amount of ethanol evaporated (mg/min) was slightly higher at the higher flow rate (3 L/min), as there was more oxygen available to carry ethanol (see Figure 5). This phenomenon is manifested in the results obtained in this study, where the concentrations of the mixture, after the three bubbling cycles, have been reduced to 98.2% and 97.8%, compared to the original concentration with 2 L/min and 3 L/min flows, respectively. The total amount of ethanol evaporated after three administrations in the 45 min of treatment was 4.64 g at the 2 L/min flow rate, and 5.38 g at the 3 L/min flow rate (see Figure 3a).

A study performed by Zhang [50] demonstrated that deposition of ethanol in several local sites of the respiratory tract using a human upper airway model depended on fluid flows and diffusion parameters. Lower flow rates allowed higher percentages of ethanol deposition in the respiratory tract than higher flows. Instead, at higher flow rates, the concentration gradient near to the wall increases alongside the mass transfer coefficient, and consequently the absorption of ethanol. This author concluded that if the objective is deposition of ethanol and not its absorption, the use of low flow rates should be recommended, due to the extended vapor residence times at low flow rates. This is in line with our approach, in which we seek ethanol deposition with minimal systemic absorption, which could produce drug interactions or adverse reactions.

On the other hand, it is convenient to highlight the safety of the current pharmacological strategy proposed. Preclinical studies in rats and mice that inhaled ethanol provide the greatest evidence available to date for characterizing the risks of inhalation of ethanol [51,52]. The observed adverse effects were attributed to systemic and chronic exposure to ethanol, regardless of the route of administration [53]. Choi et al. analyzed the efficacy and safety of the administration of proteins carried in absolute ethanol in rats. No allergic or inflammatory responses were shown, nor was damage to the alveolar barrier, or cell lysis that could indicate acute toxic effects in the lungs or airways [54]. These results are in line with what was observed in the present study after the administration of ethanol in Sprague-Dawley rats (females, 250 g (BW), with minute ventilation of 130 mL/min) every 8 hours for five days, with an inhalation period of 15 min. Bavis et al. described that rats with the same characteristics as used in our assays have a minute ventilation 0.130 L/min [55]. Taking this into account, in our experiments at 2 L/min, the rats were exposed to a total estimated amount of ethanol of 301.3 mg/day, which corresponds to a dose of 1.2 g/kg/day. At the same flow, in our study, 151 g were estimated as inhaled by human in a day. Considering a human body weight of 60 kg, the dose administrated would correspond to 0.025 g/kg/day. Thus, in the preclinical study, the exposure limits were overestimated up to 40-fold higher than those applied in the human volunteers. No significant differences were observed through histological staining by a lung pathologist specialist between the rodents that received ethanol compared to those that received control. Lung, trachea and esophagus samples were described as absent of damage.

To our knowledge, no molecular imaging studies have been published to date to characterize the biodistribution of radiolabeled ethanol administered through the respiratory tract. In order to obtain a first orientation of the possible residence time of ethanol in the lung, a PET/CT study has been carried out in rats, therefore being the first research about pulmonary kinetics of ethanol administered by this route. The PET/CT images obtained after the administration of 1-^11^C-ethanol show radioactivity at initial times at the level of respiratory tract and lungs, followed by a fast and delocalized distribution over the whole animal. These findings suggest a translocation of ethanol to the circulatory system, followed by progressive accumulation in its main organ of metabolism, the liver. These results are in line with that previously observed by Gifford et al., in which 1-^14^C-ethanol was administered intravenously in order to determine sites of concentration of ethanol or its metabolites, which may contribute to its toxicological and pharmacokinetic properties [56].

At this point, a new line of future research opens so that specialized centers can test its virucidal potential. Nowadays, our group are testing the clinical efficacy and safety of this new pharmacological strategy in 170 early-stage COVID-19 institutionalized elderly patients. This clinical trial will allow us to know if the stablished ethanol concentration is effective, if the exposure time is adequate and also the toxicity profile of inhaled ethanol. This may be possible thanks to the recent approval of the phase II clinical trial authorized by the Spanish Agency of Medicines and Medical Devices (ALCOVID-19; EudraCT: 2020-001760-29).

## 5. Conclusions

This research work aims to show the development and results of the galenic, pharmacokinetic and toxicological characterization of inhaled ethanol as a potential therapy against SARS-CoV-2. The developed 65° ethanol compounded formulation remains stable at room temperature and protected from light for 15 days. The most convenient flow rate for ethanol administration is 2 L/min, ensuring the inhalation of an ethanol daily dose of 33.6 ± 3.6 mg/min, and achieving more stable concentrations during the entire treatment (45 min). Furthermore, it has also been found to show satisfactory pharmacokinetic and toxicological characteristics through PET/CT studies and histological analysis of respiratory tract and lung tissue in rats. Clinical safety and efficacy are currently being studied in a phase II clinical trial (ALCOVID-19; EudraCT number: 2020-001760-29) for early-stage COVID-19-institutionalized patients.

## Figures and Tables

**Figure 1 pharmaceutics-13-00342-f001:**
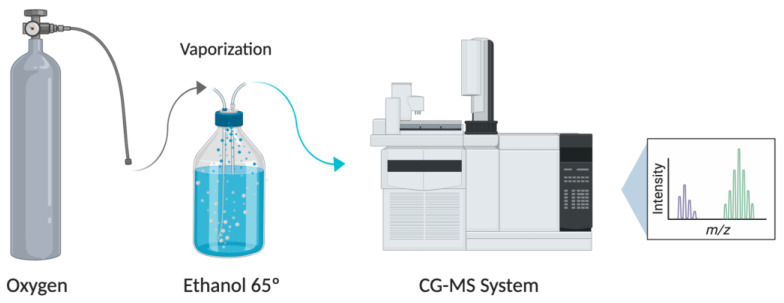
Diagram of the oxygen flow system and humidifier with 65° ethanol coupled to a gas chromatography mass spectrometry (GC-MS) system. Created with BioRender.com.

**Figure 2 pharmaceutics-13-00342-f002:**
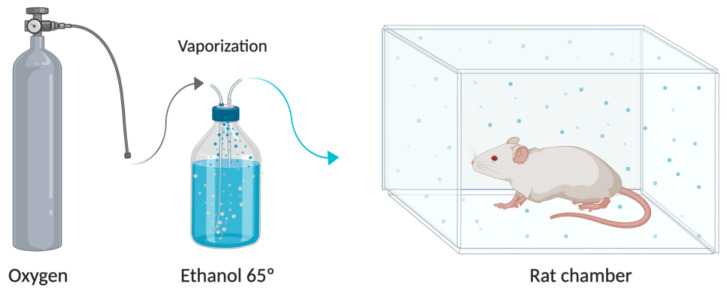
Oxygen flow system and humidifier with 65° ethanol coupled to the inhalation chamber, where the rats are exposed to ethanol vapor for inhalation. Created with BioRender.com.

**Figure 3 pharmaceutics-13-00342-f003:**
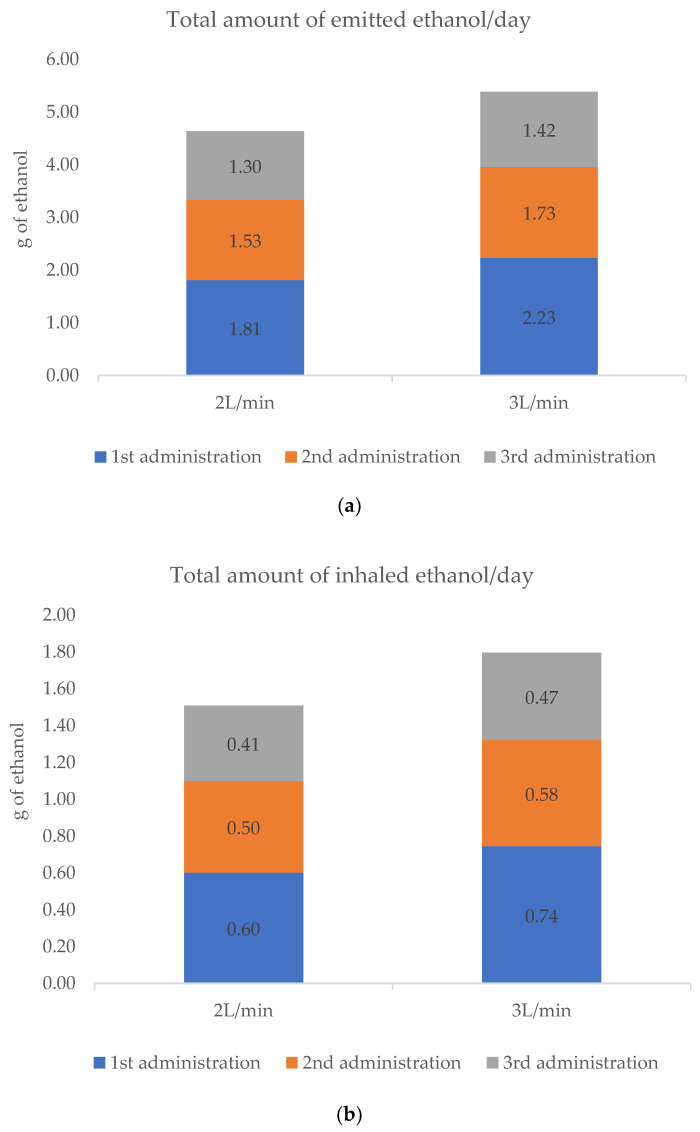
(**a**) The total amount of ethanol evaporated from the initial solution, according to GC-MS data determined by calculation of the areas under concentration-time curves (AUC), and (**b**) the estimated total amount of ethanol inhaled by human adults taking into account the recommended short-term exposure values for inhalation, in sedentary or passive activity (**b**), after the 45 min of treatment at 2 L/min and 3 L/min.

**Figure 4 pharmaceutics-13-00342-f004:**
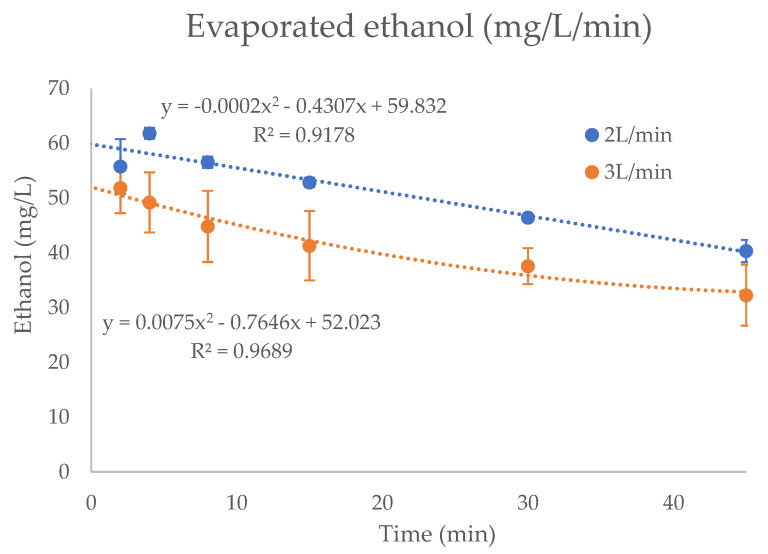
Ethanol concentration (mg/L) in the vapor phase of oxygen generated in the humidifier, during the 45 minutes of treatment, at 2 L/min and 3 L/min.

**Figure 5 pharmaceutics-13-00342-f005:**
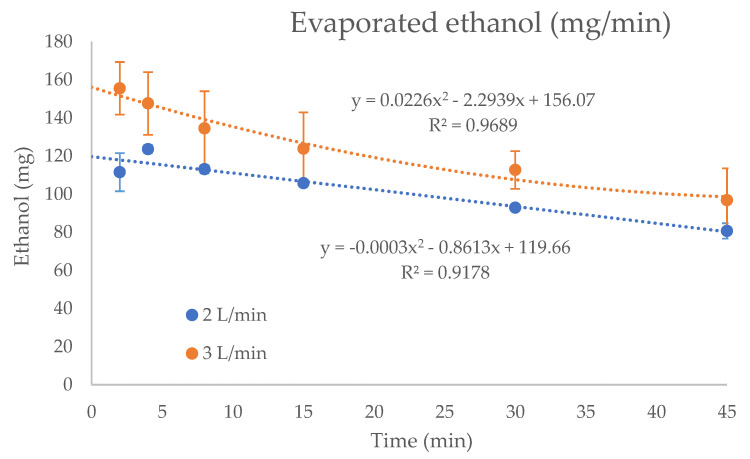
Amount of ethanol (mg) in the vapor phase of oxygen generated in the humidifier, during the 45 minutes of treatment, at the flow rates of 2 L/min and 3 L/min.

**Figure 6 pharmaceutics-13-00342-f006:**
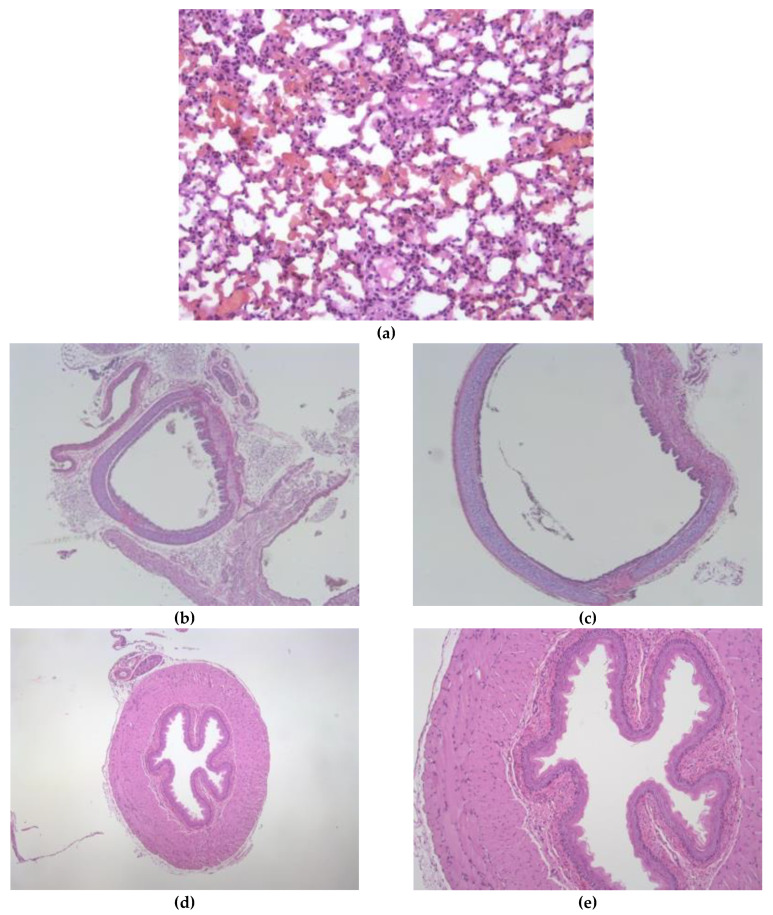
Upper respiratory tracts fixed in 10% formalin, dehydrated, paraffin embedded, sectioned in slices with 4 μm thickness and stained with H&E: (**a**) Lung (200×) with description of absence damage; (**b**,**c**) Trachea (10×, 40×) with description of absence damage; and (**d**,**e**) Esophagus (40×, 100×) with description of absence damage.

**Figure 7 pharmaceutics-13-00342-f007:**
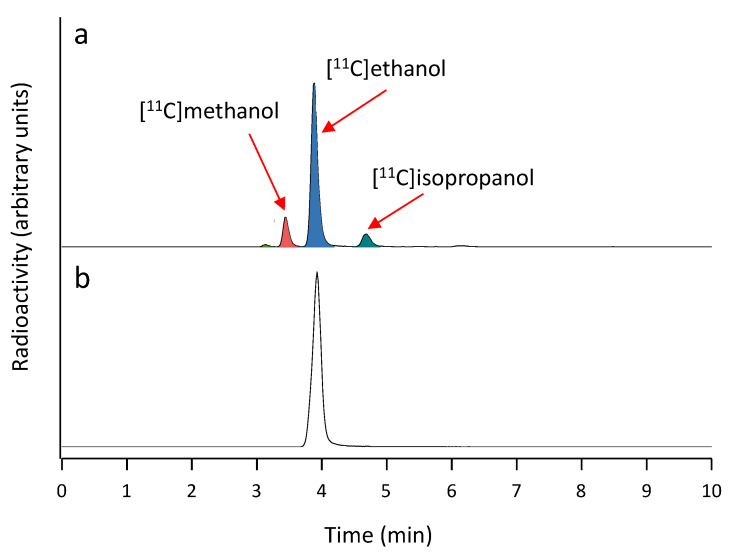
(**a**) Chromatograms (radioactivity detector) obtained after analysis of the reaction crude; (**b**) chromatograms (radioactivity detector) obtained in quality control analysis of the purified 1-^11^C-ethanol.

**Figure 8 pharmaceutics-13-00342-f008:**
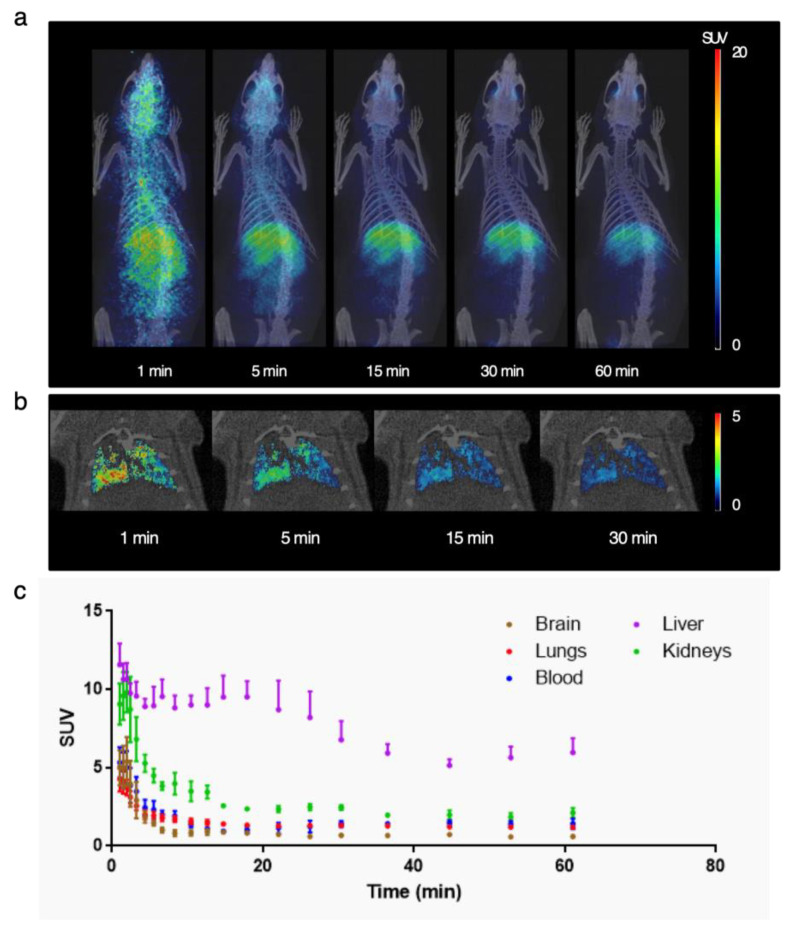
(**a**) Positron emission tomography (PET) images (maximum intensity projections, coronal views) obtained at different time points after intratracheal administration of 1-^11^C-ethanol. Images have been coregistered with 3D-rendered computed tomography (CT) images for anatomical localization of the radioactive signal; (**b**) PET images (coronal projections) corresponding to the segmented lungs at different times points after administration; images have been coregistered with representative CT slices; (**c**) time activity curves obtained after quantification of volumes of interest drawn in different organs. Values are expressed as standard uptake values (SUV), mean ± standard error mean, *n* = 3.

## Data Availability

Not applicable.

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
