# Peer review of "Development and Characterization of Inhaled Ethanol as a Novel Pharmacological Strategy Currently Evaluated in a Phase II Clinical Trial for Early-Stage SARS-CoV-2 Infection"

_pharmaceutics, 2021, doi:10.3390/pharmaceutics13030342_

Round 1

Reviewer 1 Report

Castro-Balado and colleagues’ manuscript presents a novel pharmacological strategy against SARS-CoV-2 based on the inhalation of 65° ethanol. The rationale is to decrease viremia in the first stages of the disease and, consequently, reduce the disease’s progression and improve its prognosis. The paper reports the galenic characterization of the ethanol formulation in terms of physico-chemical and microbiological stability, and the first safety investigation on both rats and healthy human volunteers. Finally, the authors also report a pharmacokinetic study in rats by PET/TC using radiolabeled ethanol. The full safety and efficacy of the formulation will be evaluated in an on-going phase-II clinical trial.

The paper is nicely written, cites appropriate references and may be of interest for the readers of Pharmaceutics. The experiments are well conducted, and the results are clearly reported and appropriately discussed. I have the following observations:

  • Authors claim in several parts of the text that the ethanol formulation has been approved, and is currently in the recruitment phase. Authors also mention the EUDRACT number of the trial. However, I am not able to find it on clinicaltrialsregister.eu. In the discussion section, authors provide a link, but it does not refer to clinicaltrialsregister.eu. Please provide the direct link of the trial in clinicaltrialsregister.eu.
  • I suggest that authors consider moving the preliminary clinical study results on healthy volunteers from the supplementary materials to the main text. They are only two pages, and they would improve the manuscript.

Minor point:

  • Line 475: please modify as “5 days”.

Author Response

Thank you very much for taking your time in reviewing our manuscript. We have written our responses after each of your comments. Changes made in the manuscript have been marked in yellow.

  1. I have the following observations: Authors claim in several parts of the text that the ethanol formulation has been approved and is currently in the recruitment phase. Authors also mention the EUDRACT number of the trial. However, I am not able to find it on clinicaltrialsregister.eu. In the discussion section, authors provide a link, but it does not refer to clinicaltrialsregister.eu. Please provide the direct link of the trial in clinicaltrialsregister.eu.

Thank you very much for the observation. In reference to the EudraCT number, we have checked it, and as you have verified, it does not appear in clinicaltrialsregister.eu. The link provided in the text (https://reec.aemps.es/reec/public/web.html) is the CLINICAL STUDIES REGISTRY of the Spanish Agency for Medicines and Healthcare Products. The information provided by this registry is also available in English. We have contacted through this agency to clarify why our clinical trial is not updated in the European registry in order to correct this error. The information will be updated as soon as possible in clinicaltrialsregister.eu.

  1. I suggest that authors consider moving the preliminary clinical study results on healthy volunteers from the supplementary materials to the main text. They are only two pages, and they would improve the manuscript.

Thank you for your appreciation. However, our idea is to present only robust results of the preclinical study in this work. We agree to publish the clinical results soon in full and not in this work as preliminary results. So far, a total of 76 patients have been recruited into the phase II clinical trial and safety results are really good.  

  1. Minor point: Line 475: please modify as “5 days”.

Thanks for the observation. We have made the suggested correction.

Reviewer 2 Report

This is a promising study describing the possible administration of ethanol (65%) for the treatment of COVOD-19. The abstract needs to be better structured, the introduction and methodology are appropriate, and the results and discussion are fine. A few points need to be addressed:

  • What are the bases for approval obtained for this formulation? Is it only safety? Is it active against covid19?
  • Why not considering the use of appropriate inhalation devices to deliver ethanol, such as nebulizers? These have been demonstrated to be appropriate for delivering ethanol in fine particle fraction in previous studies: Ghazanfari, A. Elhissi, Z. Ding, K. M. Taylor. The influence of fluid physicochemical properties on vibrating-mesh nebulization. International Journal of Pharmaceutics, 339 (2007) 103-111.
  • The authors may write their views about treatment strategies of COVID 19, alternative to vaccines (one paragraph to describe possible future research directions in this field)

Author Response

Thank you very much for your considerations, we really appreciate them. We have written our responses after each of your comments. Changes made in the text are marked in yellow.

A few points need to be addressed:

  1. What are the bases for approval obtained for this formulation? Is it only safety? Is it active against covid19?

In pandemic situations like the current one, it is advantageous to use treatments that have already been used and do not require safety studies. This is known as therapeutic repositioning, which consists on the use of a drug for another indication different than the one initially approved. On the one hand, inhaled ethanol was previously used for the treatment of acute lung edema and to avoid alcohol deprivation in postoperative neurosurgical patients. Previous studies have shown that the acute administration of inhaled ethanol for therapeutic use (in concentrations of up to 60%) has been safe, showing good tolerance and without biochemical alterations that could indicate toxic systemic exposure to ethanol. On the other hand, it has also been proved that ethanol is a molecule with activity against SARS-CoV-2. In the currently phase II clinical trial we will demonstrate its activity/efficacy when administered by inhalation for the treatment of SARS-CoV-2 respiratory infection.

  1. Why not considering the use of appropriate inhalation devices to deliver ethanol, such as nebulizers? These have been demonstrated to be appropriate for delivering ethanol in fine particle fraction in previous studies: Ghazanfari, A. Elhissi, Z. Ding, K. M. Taylor. The influence of fluid physicochemical properties on vibrating-mesh nebulization. International Journal of Pharmaceutics, 339 (2007) 103-111.

We appreciate your commentary. Indeed, nebulizers have been demonstrated to be appropriate for delivering ethanol to the respiratory tract in fine particle fractions. However, the risk of COVID transmission via droplet nuclei may be increased during nebulizer treatments because of the potential to generate a high volume of respiratory aerosols (DOI: 10.1016/j.rmed.2020.106236). These aerosols may be propelled over a longer distance than is involved in a natural dispersion pattern. In this sense, the local administration of inhaled ethanol with low oxygen flow rates does not increase the risk of spreading the disease.

  1. The authors may write their views about treatment strategies of COVID 19, alternative to vaccines (one paragraph to describe possible future research directions in this field)

Thank you for your suggestion. A paragraph regarding the most promising treatment strategies alternative to vaccines has been added in the introduction.

Reviewer 3 Report

The authors present a relatively complex manuscript transitioning from pharmaceutic development, in vitro testing, pre-cinical testing and a phase 1 trial in humans.  It is an interesting project that should be of interest to readers.  As administration of inhaled drug by vapor as opposed to aerosol, a bit of background might add value.  

Specific comments:

Line 131. "A volume of 250 mL of 65Åã ethanol solution was packaged in disposable humidifier bottles (INTL CE0482. Ref. 3230, generously donated by Oximesa Nippon Gases).

Is this the bubble humidifier used in all subsequent dosing experiments?  Please clarify.

Was any testing for leachable done with the humidifier? 

Preclinical studies:

The preclinical exposure experiments seem to be limited to standard propose dosing rather than exploring limits (like 10x exposure)  that might better support safety.

The PET study was a great interest, but appeared to be aerosol rather than vapor administration.  

Line 208 - what was the estimated inhaled dose for the rats with the chamber used?

Line 262 For the PET study "The labelled compound (ca. 7 MBq, 100 μL) was 262administered through endotracheal insufflations using the PennCentury MicroSprayer."  Please clarify whether the drug was administered as aerosol for these tests.  

If so, how do authors reconcile difference between inhaled dose by aerosol vs Vapor.

Line 293 The authors characterize the amount of vapor emitted from their system over time.  "The difference between flow rates was 0.75 g of ethanol inhalation..."  However, there are no calculations or rationale provided as to how emitted vapor contents relates to inhaled dose.  

Figure 3 is title "Total amount of inhaled ethanol/day".  

It appears this should be the amount of "emitted" ethanol/day.  Authors should differentiate between inhaled and emitted.

The authors should provide an estimate of how vapor content in 2 or 3 L/min of gas relates to inhaled dose of both rats and humans.

As example: a typical adult with 10 L/min minute ventilation with an Insp to Expiratory ratio of 1:2 would inhale a low precentage <33%) of the vapor containing gas administered.  

Discussion:  The first 3 paragraphs do not directly relate to the study done.  This seems to be more relevant to the introduction.   

The phase 1 trial in the appendix Figure 1a shows Ethanol 65 administered through a jet nebulizer attached to a Ventimask.  

What is the role of the nebulizer in the circuit?  2 - 3 L/min is insufficient to drive most jet nebulizer (6 - 10 L/min typically operating flows).  The mask appears to have valves on the mask ports.  This would be dangerous for administration of such low flows, as it would be obstruct patient breathing.  

Why did authors not use simple nasal cannula?  Cannulas have been shown to administer as much or more low flow oxygen as simple aerosol masks.

How would this system be used with patients requiring higher flows of oxygen?  

Round 2

Reviewer 3 Report

The author response has clarified and improved the manuscript